# EFFICIENT GENERATION OF DIVERSE SCIENTIFIC HYPOTHESES THROUGH STEPWISE CONCEPTUAL CONCRETIZATION

## ABSTRACT

In recent years, the automation of research using LLMs has been advancing rapidly. While The AI Scientist can generate papers that meet the acceptance criteria of top conferences in the machine learning field under specific conditions, there are limitations to the innovativeness of the generated research. As a step toward improving quality, this study aims to develop a method that generates scientific hypotheses of equivalent quality with significantly fewer tokens. The proposed method, which generates hypotheses more than ten times more efficiently, was compared with previous research in terms of novelty, singnificance, clarity, feasibility, and validity of the generated hypotheses. While no clear differences were observed in novelty and feasibility, improvements in performance were recognized in terms of singnificance, clarity, and validity compared to previous research.

## 1 INTRODUCTION

Over the past years, automation technology has been rapidly introduced in scientific research, with particular attention being paid to research automation utilizing LLMs (Baek et al., 2024; Ifargan et al., 2024; Zhou et al., 2024; Ghafarollahi & Buehler, 2024; Qi et al., 2023; Rives et al., 2021). The AI Scientist (Lu et al., 2024) aims to fully automate everything from research idea generation to experiment implementation and execution to paper writing, demonstrating its ability to generate papers in the field of machine learning that receive automatic peer review results equivalent to top conference acceptance.

However, there are limitations to the innovation of the generated research, and achieving higher quality scientific discoveries remains a challenge. According to research findings by Large Language Monkeys, it has become clear that in domains such as programming, solution quality improves as the number of generated samples increases (Brown et al., 2024). However, The AI Scientist's hypothesis generation method is computationally expensive, making it difficult to apply a large-scale trial approach.

This research aims to develop a method that generates scientific hypotheses of quality equivalent to The AI Scientist using significantly fewer tokens. For evaluation, we will conduct multifaceted comparisons with The AI Scientist and perform ablation experiments to verify the effectiveness of each component of the proposed method.

## 2 METHOD

### 2.1 PIPELINE: TREE OF GENERATION

While previous research generated hypotheses one at a time, requiring previous hypotheses to be re-input each time, the proposed Tree of Generation method is characterized by multi-step generation of multiple ideas that progressively concretize concepts. This enables efficient enhancement of diversity while reducing the required number of tokens by an order of magnitude.

More specifically, Tree of Generation (Figure 1) uses a Large Language Model (LLM) to: (i) first generate `NUM_ITEMS` mechanisms related to the `field`, (ii) generate the same number of experimental ideas using these mechanisms, and (iii) further generate the same number of more detailed experimental procedures. Additionally, during step (iii), it performs a self-evaluation of interestingness on a 10-point scale and filters based on a threshold.

```
(i) Generate {NUM_ITEMS} distinct mechanisms related to
{field} research.

(ii) Generate {NUM_ITEMS} distinct experimental designs
to test the following mechanism: "{mechanism}"

(iii) Based on this experimental concept: "{experiment}"
Generate {NUM_ITEMS} more specific and concrete versions
of this experiment with precise methodological details.
```

Figure 1: Excerpt of the main prompt sections

Regarding computational complexity, while previous research (blue) requires input tokens to increase quadratically with the number of hypotheses generated, Tree of Generation (red) keeps it to the order of two-thirds power (Figure 2).

For experimental settings, we adopted a temperature of 0.7, `NUM_ITEMS` of 7, LLM as the input field, an interestingness threshold of 10 or higher, and gpt-4o-2024-05-13 (Hurst et al., 2024) as the large language model. Data from previous research uses the data published by The AI Scientist authors on GitHub (`https://github.com/SakanaAI/AI-Scientist`). Additionally, to ensure comparable conditions, we filtered using the same interestingness threshold and sampled to achieve the same numbers.

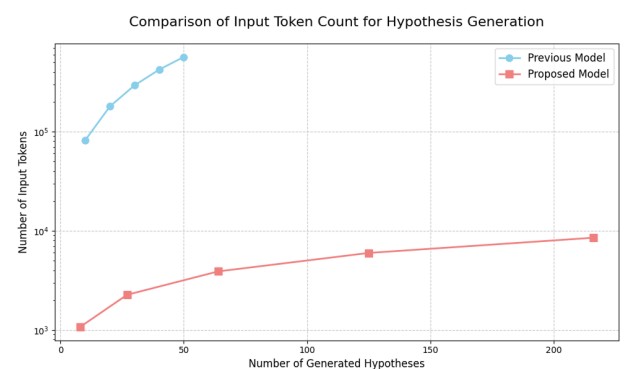

Figure 2: Relationship between number of hypotheses and input tokens

## 2.2 Evalutaion Method

For the evaluation of this research, we primarily use Idea Arena (Li et al., 2024). This is a type of qualitative evaluation using LLMs (LLM-as-a-judge), where for all possible pairs of hypotheses, superiority, inferiority, or a draw is determined for each evaluation axis. Winners receive 1 point, losers receive 0 points, and in case of a draw, each receives 0.5 points per evaluation axis, with results being mechanically aggregated. To avoid bias from the order of prompts, we present hypothesis A followed by hypothesis B to the LLM, then present them in reverse order, conducting two-time evaluations per pair. The model used for evaluation is the same as the proposed method, gpt-4o-2024-05-13, with temperature set to 0.

## 3 Results

### 3.1 Comparison with Previous Research

Figure 3 compares the novelty, singnificance, clarity, feasibility, and effectiveness of the generated hypotheses with The AI Scientist, visualized using kernel density estimation. The proposed method, shown in blue-green, achieved equal or higher evaluations across all assessment criteria compared to previous research.

Among the results, the most significant differences were found in singnificance, clarity, and effectiveness. Singnificance scored 19.2 points higher on average compared to previous research, clarity was 8 points higher, and effectiveness was 15.7 points higher. The standard deviations were 5.71 for

singnificance in previous research versus 5.37 for the proposed method, 2.08 versus 3.52 for clarity, and 7.21 versus 3.71 for effectiveness, with the differences in means all exceeding 2 standard deviations.

For other metrics, novelty and feasibility showed no clear differences. In terms of novelty, while the standard deviations were 4.36 for previous research and 8.44 for the proposed method, the difference in means was only 0.88. For feasibility, the standard deviations were 3.90 and 2.68, with a mean difference of 1.76.

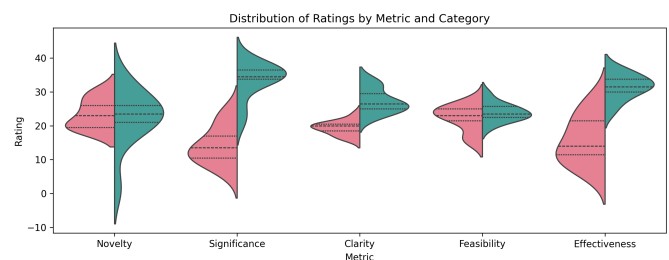

Figure 3: Comparison with previous research across five axes. the red is the previous method and the blue-green is the proposed.

These results indicate that this study's goal of generating hypotheses of similar quality more efficiently has been achieved. Additionally, reproducibility has been confirmed through multiple sets of trials. While there is some variation, generally equivalent hypotheses can be generated (Figure 6).

However, caution is needed in interpreting these results. While the previous method generated hypotheses in fields such as diffusion models (Croitoru et al., 2023), nanoGPT (Karpathy, 2022), and grokking (Power et al., 2022), this method generated LLMs. It's possible that the broader range of fields could have contributed to higher novelty scores. On the other hand, the baseline data obtained from previous research spans multiple fields, which could be considered as covering a broad range of areas when combined.

## 3.2 ABLATION OF EACH STEP

To measure the impact of each of the three steps that constitute the Tree of Generation on performance, Figure 4 compares the performance when any of the three steps of the proposed method is removed against the original Tree of Generation (red). A clear decrease in performance is observed in all cases, indicating that all steps contribute to the performance improvement of the Tree of Generation.

When excluding the first step, (i) enumeration of mechanisms (blue-green), performance decreases to some extent, particularly in novelty, singnificance, and effectiveness. Novelty shows an average decrease of 10.4 points, singnificance decreases by 9.3 points, and effectiveness shows a decrease of 23.8 points. When removing the second step, (ii) enumeration of experimental ideas (purple), novelty and singnificance decrease significantly, and effectiveness also shows a relatively large decrease. Novelty shows an average decrease of 34.8 points, singnificance decreases by 25.9 points, and effectiveness shows a decrease of 18.3 points. When removing the third step, (iii) enumeration of experimental processes (ochre), clarity, feasibility, and effec-

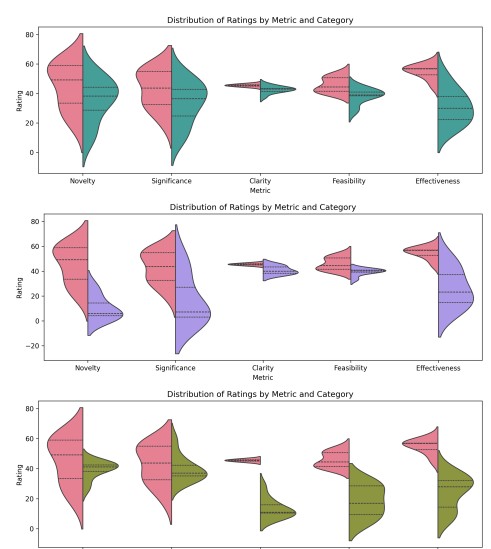

Figure 4: Ablation by generation step. The red is the proposed method, the blue-green is removing the first step, the purple is excluding the second step, and the ochre is removing the third step.

tiveness decrease significantly. Clarity shows
an average decrease of 32.1 points, feasibility decreases by 27.4 points, and effectiveness shows a
decrease of 30.4 points.

Summarizing these results, step (ii), which corresponds to thinking at a more abstract level about
experiments, particularly enhances novelty and singnificance, while step (iii), which corresponds to
thinking at a more concrete level, particularly enhances clarity and feasibility. The former result
is understandable if we consider that the research perspective influences novelty and singnificance.
The latter is also easy to understand if we consider that specific experimental processes affect these
evaluation metrics. Additionally, if we consider the first step as corresponding to the concretiza-
tion of research themes, it explains why it contributes to performance improvement across various
aspects.

### 3.3 TUNING OF OUTPUT NUMBERS

Here, we compare the results by varying
`NUM_ITEMS`. The red area represents when
`NUM_ITEMS` is set to 7, the green is 3, and the
light blue is 5. Compared to when it is 7, we can
see that novelty, singnificance, and effective-
ness substantial decrease in both cases. In par-
ticular, the smaller the number, the greater the
decrease, demonstrating that generating many
items at each step is crucial for the proposed
method.

### 3.4 BEHAVIOR
OF INTERMEDIATE OUTPUTS

Next, we analyze the semantic changes in out-
puts at each step. Table 1 shows the cosine dis-
tances (i.e., the difference between 1 and co-
sine similarity) between the embeddings gen-
erated at each step and the given field, as well
as the cosine distances with the previous step.
The embeddings were generated using OpenAI
API's text-embedding-3-large.

Each row shows the mean or
standard deviation. The first and
second rows use the cosine dis-
tances between the generated re-
sults at each step and the field.
The third and fourth rows use
the cosine distances between the
generated results at each step and
the previous step. For example,
in column (iii), it shows the dis-
tance from (ii) which was the

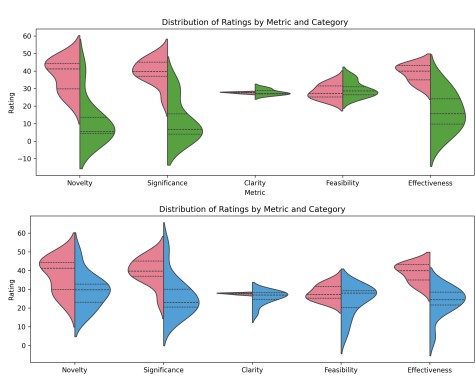

Figure 5: Adjusting output numbers. The red is
set to 7, the green is 3, and the light blue is 5.

Table 1: Cosine distances between generated outputs and fields,
or cosine distances from previous steps.

| metrics | (i) | (ii) | (iii) | (iii) only |
|---|---|---|---|---|
| mean (vs. field) | 0.700 | 0.862 | 0.851 | 0.545 |
| s.d. (vs. field) | 0.025 | 0.048 | 0.110 | 0.085 |
| mean (vs. prev.) | N/A | 0.640 | 0.560 | N/A |
| s.d. (vs. prev.) | N/A | 0.105 | 0.155 | N/A |

source of generation. Additionally, in (i), the fourth row is omitted as it would be the same as
the second row. This is also true for the third and first rows.

The rightmost column shows the case where only step (iii) was generated, excluding (i) and (ii).
Compared to this, the adjacent (iii) shows higher means and standard deviations. This suggests that
multiple steps are necessary to increase the diversity of outputs.

Looking at the standard deviation of distances from the field (second row), it increases as steps
progress, suggesting an increase in diversity. This aligns with the results in the fourth row. Regarding
the mean distance from the field (first row), it slightly decreases from (ii) to (iii). This might be
interpreted as pulling back the expanded ideas through concretization. The mean distance from the

previous step (third row) decreases with each step. This suggests that concepts generated in earlier steps are more distant, while those in later steps are closer, which is consistent with the step-by-step concretization approach.

Combined with the ablation results from Section 4.2, it appears that (ii) enhances novelty and singnificance by increasing diversity in terms of mean distance at this step. This is supported by the fact that the mean distance from the field is greatest at this step. While the mean distance from the previous step is larger in (i), this step merely lists mechanisms and presumably doesn't enhance novelty and singnificance as much as (ii).

The enhancement of clarity and feasibility in (iii) seems to be achieved by maintaining rather than increasing semantic spread. This is suggested by the mean distance from the field not increasing beyond that of (ii). Furthermore, the increased standard deviation in distances from both the field and the previous step can be interpreted as a result of enhanced clarity and feasibility. Improving these aspects doesn't allow for rough content, and this constraint likely creates variations in distances, leading to larger standard deviations.

## 3.5 QUALITATIVE EVALUATION

The prior method proposes integrating a mesh convolutional network (MeshCNN; Hanocka et al., 2019) into a diffusion model (Croitoru et al., 2023), while the Tree of Generation aims to evaluate adversarial robustness in a federated learning environment (Kairouz et al., 2021). More specifically, the former (Figure 7) incorporates a new mechanism called MeshCNN into the diffusion model to enable deeper understanding of data structures. It treats 2D data as a mesh (a network composed of vertices and triangles) and utilizes its characteristics to improve the quality of generated data. Its effectiveness is evaluated from the perspectives of data generation quality and computational efficiency. The latter (Figure 8) focuses on federated learning, a mechanism where multiple distributed clients cooperate to train a machine learning model, and aims to evaluate resilience against malicious attacks. Using certain data, each client conducts training to enhance attack resistance, after which the overall model is integrated to measure how well robustness is maintained.

The former exhibits high novelty in that it integrates MeshCNN, which is typically used for geometric data like 3D models and meshes, into a 2D data diffusion model. The latter shows high novelty in bringing adversarial methods, typically used in non-distributed environments, into the distributed context of federated learning. The Tree of Generation appears to achieve a similar level of novelty as prior research.

Regarding singnificance, while the former could lead to performance improvements in fields requiring point cloud processing (CAD, robotics, medical imaging, etc.), its impact is somewhat indirect. The latter directly addresses critical challenges in safely operating federated learning in real-world environments.

In terms of clarity, both follow a step-by-step explanation and are comprehensible given the reader's knowledge base. If anything, the Tree of Generation's structured approach might be slightly more readable, but the content appears to be at a similar level.

Regarding feasibility, while both approaches tend to increase computational complexity - the former with MeshCNN and the latter with adversarial methods - they can be implemented using existing datasets and libraries. The Tree of Generation seems to demonstrate similar feasibility to prior research.

It should be noted that the Tree of Generation's hypothesis lacks some specificity in parts. It doesn't specify which model to adopt for federated learning. However, the prior research hypothesis also doesn't specify which model to use specifically in the diffusion model. Considering these factors, it appears to be generating hypotheses at a similar level.

## ACKNOWLEDGMENTS

We thank the anonymous reviewers for their comments.

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

## A  RELATED WORKS

### A.1  BASELINE OF THE PROPOSED METHOD

This research uses The AI Scientist (Lu et al., 2024) as its baseline. This is a framework that automates the entire process from research idea generation to paper writing and peer review evaluation using large language models. The AI Scientist's processing flow can iteratively execute a series of processes including research idea generation, code implementation, experiment execution, result visualization, paper writing, and peer review evaluation. A distinctive feature is its hypothesis generation based on iteratively feeding examples of previously generated hypotheses back into the input and exploring thought processes based on the output.

### A.2  AUTOMATIC HYPOTHESIS GENERATION USING LLMS

Research on automatic hypothesis generation using LLMs can be classified into three approaches based on their information sources. The first approach uses knowledge graphs to perform structured reasoning, with proposals such as KG-CoI (Xiong et al., 2024) and SciAgents (Ghafarollahi & Buehler, 2024). The second approach generates and updates hypotheses based on experimental and observational data, including methods that apply multi-armed bandit concepts (Zhou et al., 2024) and data-to-paper (Ifargan et al., 2024). The third approach generates new research ideas from scientific literature, with proposals such as ResearchAgent (Baek et al., 2024) and VirSci (Su et al., 2024). However, many of these assume large amounts of input information, and methods for generating hypotheses without using new external knowledge have not yet been sufficiently studied.

### A.3  EVALUATION METHODS FOR GENERATED HYPOTHESES

There are two approaches to evaluating generated hypotheses: human evaluation and LLM evaluation. Human evaluation has constraints in terms of cost and time, and evaluator bias tends to have a greater impact. On the other hand, LLM evaluation includes proposals such as absolute evaluation by ReviewingAgents (Baek et al., 2024) and relative evaluation by Idea Arena Li et al. (2024). Idea Arena, in particular, enables the detection of finer differences by relatively comparing multiple ideas. This research adopts the relative evaluation approach of Idea Arena due to its advantage in detecting subtle differences between methods.

## B  ADDITIONAL EXPLANATION FOR PROPOSED METHOD

This simple proposed method works because it extracts carefully selected hypotheses from among many that have efficiently increased diversity. As mentioned above, diversity is effectively enhanced through the three-step generation process and by setting the LLM's temperature higher to introduce randomicity (see Section 4.2 for visualization of generation result diversity). Furthermore, as the second and third generations progress, the specificity increases, meaning the generated text volume grows and the number of usable concepts increases, which is expected to significantly reduce the possibility of duplication.

After increasing diversity in this way, many hypotheses are generated and then carefully selected. Since only the cream of the crop will ultimately bear fruit as research outcomes, it's not problematic that some lower-quality hypotheses are included among the many generated in the intermediate steps. This design maintains the quality of ultimately adopted hypotheses while reducing the number of input tokens.

(This space left intentionally blank.)

## C    MAIN RESULT REPRODUCTION

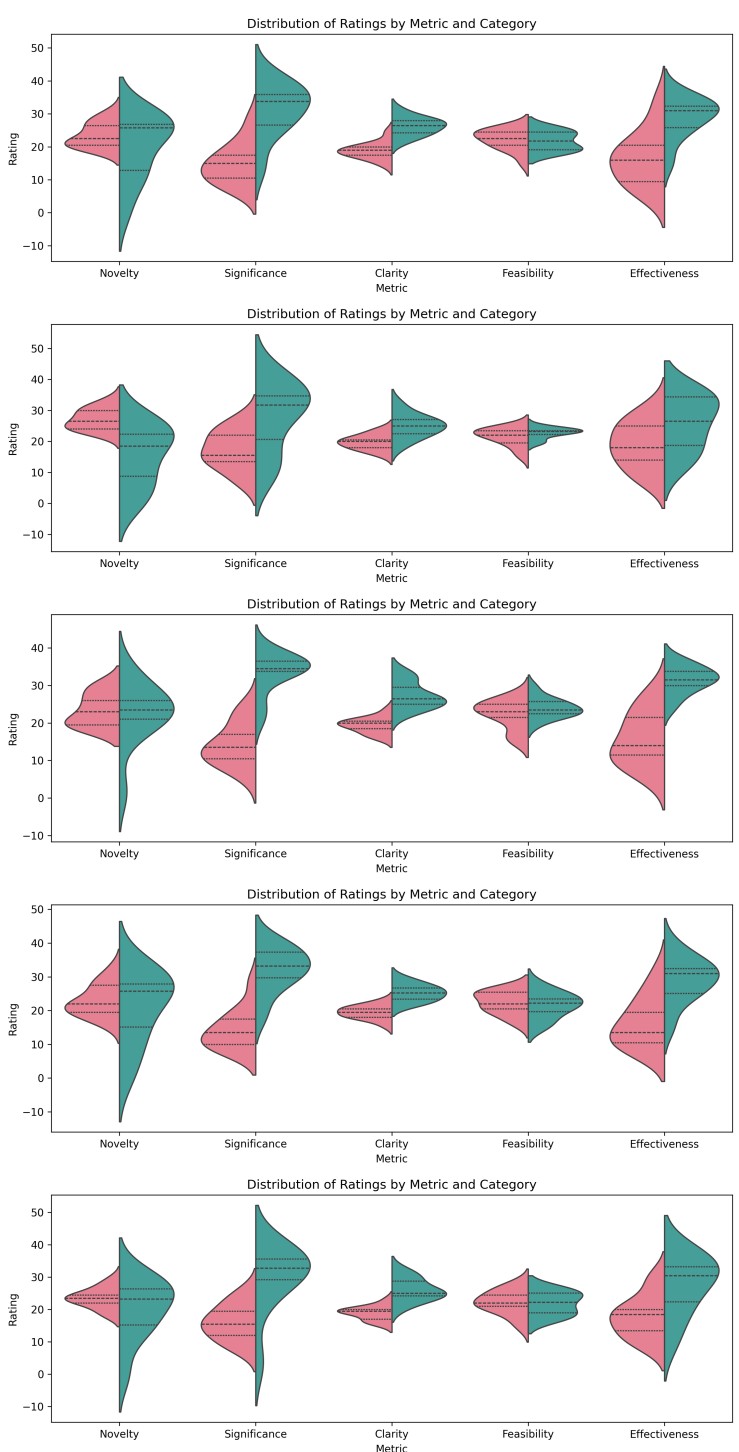

Figure 6: Multiple comparison with prior research

## D  OUTPUTS FOR QUALITATIVE EVALUATION

Enhancing Diffusion Models with Mesh Convolutional Networks for
Geometric Learning

In this experiment, we will integrate a Mesh Convolutional Network
(MeshCNN) into the diffusion model. Specifically, we will: (1) Implement
a new MeshCNNEmbedding class that uses MeshCNN to generate embeddings
from the input data, (2) Construct a mesh from the 2D data points by
treating them as vertices and using Delaunay triangulation for
connectivity, (3) Modify the MLPDenoiser to use these MeshCNN embeddings
along with the existing positional and temporal embeddings by
concatenating them together, (4) Adjust the training loop to incorporate
the new embeddings, and (5) Train the modified model on the same
datasets. We will compare the results in terms of training time,
evaluation loss, KL divergence, and sample quality using both
quantitative metrics and qualitative visual inspection. The impact of
the MeshCNN embeddings will be evaluated through metrics such as FID and
visual quality of the samples.

Figure 7: Hypothesis generated by prior research

Adversarial Robustness in Federated Learning

Objective: Assess the generalization of adversarial robustness in a
federated learning setting.
Procedures:
  - Simulate a federated learning environment with multiple clients
    using different datasets (e.g., CIFAR-10, MNIST).
  - Train a global model using FGSM for adversarial training at each
    client.
  - Test the aggregated global model with adversarial examples from PGD
    and CW attacks.
  - Measure model performance on clean and adversarial examples.
Evaluation:
  - Compare robustness metrics across different clients and the global
    model.
  - Success is defined by maintaining high accuracy and robustness in
    the federated setting.

Figure 8: Hypothesis generated by Tree of Generation

