# OpenReview forum: "Efficient Generation of Diverse Scientific Hypotheses through Stepwise Conceptual Concretization"
_ICLR.cc/2025/Workshop/AgenticAI — ICLR 2025 Workshop AgenticAI Reject_

### Official Review · Reviewer_5Sxu · 2025-03-03
**An effective method, but with incremental innovation and possibly limited applicability**

**Rating:** 5
**Confidence:** 4

**Review:**

This paper proposes the Tree of Generation, a stepwise method to efficiently generate diverse scientific hypotheses while reducing computational costs. By progressively refining ideas through multiple stages, it improves significance, clarity, and effectiveness compared to previous method The AI Scientist, while maintaining similar novelty and feasibility.

Strengths:
1. This paper proposes a simple and easy-to-use method to generate scientific hypotheses. Empirical results verified the quality of the generated hypotheses.

2. The proposed method requires significantly fewer input tokens for hypotheses generation.

Weaknesses:
1. [Motivation] Through out this paper, neither is it clearly discussed why it is needed to use fewer tokens, nor is it supported by empirical results that fewer tokens will bring significant benefits.

2. [Effectiveness] This paper claims to increase "diversity" of generated hypotheses. However, all related analysis is based on the functionality of the model itself, without comparing it to state-of-the-art model. Besides, all analysis is intuitive, except for Table 1, which cannot provides convincing empirical evidence.

3. [Limited Applicability] The way that the proposed model works is to generate multiple hypotheses according to a pre-defined three-step template. However, it is not discussed whether this template can cover the majority of possible hypotheses generation problems. Without such a template, the LLMs themselves are able to process different hypotheses generation problems. This feature of the method may lead to limited applicability.

4. [Clearity] In Figure 2, the number of tokens needed for different models are compared. It is not clear defined how the "number of generated hypotheses" is counted. It would be unfair if the proposed model generates 10 hypotheses in a batch using the same command while the compared baseline generate 10 different hypotheses using different commands.

5. [Typo?] In this paper, "section 4.2" is mentioned twice, while there is no such section. I reckon this may be a typo.

---

### Official Review · Reviewer_4nyi · 2025-03-05

**Rating:** 4
**Confidence:** 4

**Review:**

This paper introduces a method called Tree of Generation, designed to efficiently generate diverse scientific hypotheses using fewer computational resources than existing AI-driven research automation methods like The AI Scientist. Instead of generating hypotheses one at a time, the proposed method structures the generation process in multiple steps, progressively refining ideas through conceptual concretization. This approach significantly reduces token usage while maintaining or improving hypothesis significance, clarity, and validity. The method is evaluated using LLM-based comparative analysis, showing that it produces hypotheses of similar novelty and feasibility but with higher clarity and effectiveness compared to previous approaches.

Strengths of the Paper:
1. The proposed approach produces more refined and structured hypotheses by progressively concretizing concepts.
2. The proposed approach can be applied across multiple scientific fields, increasing its versatility.

Weaknesses of the Paper:
1. Evaluation is not human-verified, raising concerns about real-world applicability.
2. No major improvement in novelty or feasibility, limiting its impact on groundbreaking discoveries.

---

### Decision · Program_Chairs · 2025-03-05

Reject